# Risk of systemic lupus erythematosus flare after COVID-19 hospitalization: A matched cohort study

Arthur Mageau[1,2,3], Christel Géradin[2,4], Kankoé Sallah[5,6], Thomas Papo[2,3], Karim Sacre[2,3‡]*, Jean-François Timsit[1,7‡]

1 IAME, UMR 1137 INSERM, Team Descid Université Paris Cité and Université Sorbonne Paris Nord, Paris, France, 2 Département de médecine interne, Hôpital Bichat-Claude Bernard, Assistance Publique- Hôpitaux de Paris, Université Paris Cité, Paris, France, 3 CRI, UMR 1149 INSERM, ERL 8252 CNRS, LabEx Inflamex, Université Paris Cité, Paris, France, 4 Sorbonne Université, Inserm, Institut Pierre-Louis d'Epidémiologie et de Santé Publique, Paris, France, 5 INSERM CIC-EC 1425, Hôpital Bichat Claude Bernard, Paris, France, 6 Clinical Research, Biostatistics and Epidemiology Department, AP-HP Nord-Université Paris, Paris, France, 7 Département de Réanimation Médicale et Infectieuse, Assistance Publique- Hôpitaux de Paris, Université Paris Cité, Hôpital Bichat-Claude-Bernard, Paris, France

‡ KS and JFT are joint senior authors on this work.
* karim.sacre@aphp.fr

**Data Availability Statement:** The datasets analyzed during the current study are not publicly available due the confidentiality of data from patient records, even after de-identification. However, access to the AP-HP data warehouse's raw data

## Abstract

### Objective

To analyze the risk of systemic lupus erythematosus (SLE) flare after admission for COVID-19.

### Methods

We performed a matched cohort study using the Assistance Publique—Hôpitaux de Paris Clinical Data Warehouse which collects structured medical, biological and administrative information from 11 million patients in Paris area, France. Each SLE patient hospitalized with a COVID-19 diagnosis code between March 2020 and December 2021 was matched to one SLE control patient with an exact matching procedure using age ±3 years, gender, chronic kidney disease, end-stage renal disease, and serological activity. The main outcome was a lupus flare during the 6 months follow-up. A flare was considered if a) documented by the treating physician in the patient's EHR and b) justifying a change in SLE treatment. The electronic health records (EHRs) were individually checked for data accuracy.

### Results

Among 4,533 SLE patients retrieved from the database, 81 (2.8%) have been admitted for COVID-19 between March 2020 and December 31, 2021, and 79 (n = 79/81,97.5%) were matched to a unique unexposed SLE. During follow-up, a flare occurred in 14 (17.7%) patients from the COVID-19 group as compared to 5 (6.3%) in the unexposed control group, including 4 lupus nephritis in the exposed group and 1 in the control group. After adjusting

can be granted following the process described on its website: www.eds.aphp.fr, contacting the Ethical and Scientific Committee at secretariat. cse@aphp.fr. A prior validation of the access by the local institutional review board is required. In the case of non-APHP researchers, the signature of a collaboration contract is mandatory.

**Funding:** Arthur Mageau was supported by a PhD fellowship provided by Fondation pour la Recherche Médicale (FDM202106013488). This work was supported by the Agence Nationale de la Recherche (n˚ANR-21-COVR-0034 COVALUS), the University Paris Cité and the Assistance Publique Hôpitaux de Paris The funders had no role in study design, data collection and analysis, decision to publish, or preparation of the manuscript.

**Competing interests:** The authors declare no competing interests.

for HCQ use at index date and history of lupus nephritis, the risk of flare was higher in exposed SLE patients (hazard ratio [95% confidence interval] of 3.79 [1.49–9.65]).

## Conclusions

COVID-19 hospitalization is associated with an increased risk of flare in SLE.

## Introduction

The SARS-CoV2 pandemic reopened the unresolved question of whether and how a viral infection can trigger flares of immune-mediated inflammatory diseases such as systemic lupus erythematosus (SLE). Different elements may contribute to a higher risk of lupus flare following COVID-19. First, type 1-interferon and neutralizing anti-type I interferon (IFN-I) autoantibodies—both involved in SLE and COVID-19– have a Janus effect on the activity of SLE and the control of SARS-CoV-2 infection: a high titer of neutralizing anti- IFN-I protects against lupus flare-ups but predisposes to severe COVID-19 [1–5]. In such setting, strong IFN-I production induced by SARS-CoV-2 may overpass the protective role of anti-I IFN-I in SLE and contribute to lupus flare as previously reported [6]. Second, severe COVID-19 is associated with a poor prognosis among patients with SLE [7, 8]. Tapering immunosuppressive (IS) drugs because of ongoing severe COVID-19 may increase the risk of SLE flare. So far SLE flares following COVID-19 have been reported in case reports [6, 9–11] but such association has not been assessed through larger systematic studies.

Assistance Publique–Hôpitaux de Paris (APHP) is the largest university hospital system in Europe and sees more than 8.3 million patients per year in 39 hospitals in Paris and its surroundings. We herein leverage the APHP Clinical Data Warehouse–a population-based register combined with clinical databases built from electronic health records—to collect information on SLE patients admitted for COVID-19 and perform a matched cohort study investigating the relationship between COVID-19 and lupus flare onset.

## Methods

### Data source

The Assistance Publique—Hôpitaux de Paris (APHP) Clinical Data Warehouse routinely collects and aggregates on a daily basis all the anonymized data generated through hospitalizations and consultations in a group of 39 public hospitals located in the Greater Paris area, France. The database gathers structured medical, biological and administrative information prospectively collected from 11 million patients. Diagnoses and procedures identified during the hospital stays are coded according to the International Classification of Diseases, 10th Revision [12] (ICD-10) and the *classification commune des actes médicaux* [13] (CCAM), respectively. In addition to data identified through ICD/CCAM codes, APHP Clinical Data Warehouse gives access to all the anonymized medical reports and prescriptions written in natural language in the electronic health records (EHRs) as well as all the medical notes and biological exams performed overtime during routine care.

### Study population and definitions

All patients older than 16 years of age admitted between July 15, 2017 and February 9, 2022 in any of the 39 French public hospitals located in the greater Paris area who received at least one

international classification of diseases, M32.X ICD-10 code (SLE) were retrieved from the database. Among them, 1) all SLE patients admitted for a first COVID-19 before January 1st, 2022 defined the "exposed group" and 2) all matched SLE patients without evidence for a COVID-19 prior the index date defined the "unexposed group". The follow-up of each exposed patient started the day of admission for COVID-19 (index date). Follow-up of each unexposed patient began on the index date of his or her matched exposed counterpart.

The initial selection of patients was made using M32.X and U07.1 ICD-10 codes for SLE and COVID-19, respectively. The medical charts of the selected patients were next individually reviewed to confirm that i) all patients fulfilled the 2019 EULAR/ACR classification criteria for SLE [14], ii) all SLE patients identified with COVID-19 (exposed) had a proven infection (i.e. positive PCR for SARS-CoV-2), iii) COVID-19 was the main reason for admission of exposed patients and iv) all SLE patients identified without COVID-19 (unexposed) had no evidence for any COVID-19 before the index date. The medical charts were reviewed by a physician specialized in clinical immunology (AM). Data were accessed on February 9, 2022. Authors had not any access to information that could identify individual participants during or after data collection.

Demographic characteristics, comorbidities, anti-SARS-CoV2 vaccine status, SLE features including serological activity, and treatment received prior and during index date were retrieved from medical records. Serum C3 levels and anti-dsDNA IgG titers measured within 6 months prior to the index visit were considered for serological activity. When performed, serological activity was defined as either normal—when both C3 level and anti-dsDNA IgG titer were into the normal range–or abnormal–when C3 level was low and/or anti-dsDNA IgG titer was high. A SLE flare was defined if was 1) considered by the physician in the medical record and ii) followed by an increase in dose or number of medications given for SLE in the setting of care.

## Matching procedure

Each SLE patient hospitalized because of a symptomtic COVID-19 episode during the period of study was randomly matched with one SLE control patient. A random exact matching procedure (without replacement) using the prespecified following matching variables: age ±3 years, gender, chronic kidney disease, end-stage renal disease (chronic dialysis or renal transplant), and serological activity was used. Matching accuracy and efficacy were estimated by calculating the absolute mean standardized differences (SMD) between the characteristics of the matched populations.

## Survival analyses

The primary outcome was the occurrence of a SLE flare during follow-up. Follow-up began the first day of the index date and ended after 6 months, another COVID-19-episode, death or lost to follow-up. Kaplan-Meier method was used to analyze the 6-months survival without flare. We used a marginal Cox proportional hazard model [15], accounting for the matching, to calculate the hazard ratio of the exposed population, using the unexposed population as reference. We adjusted the model on the remaining relevant differences between the two matched populations (i.e. use of hydroxychloroquine at index date and a history of a lupus nephritis). In order to look for factors associated with the occurrence of flare in the exposed group while taking account of the high mortality rate in this population, we ran univariate Fine and Gray's competing risk models [16] using either occurrence of flare or death during follow-up as outcomes of interest.

## Statistical analyses

Categorical variables are given as number (percentage). Quantitative variables are given as median (first quartile–third quartile). HRs are given with their 95% CI. The evolution of serological activity—C3 levels in all and anti-dsDNA IgG titers among in those with positive anti-dsDNA IgG—following index date were fitted and plotted with a LOESS approach. We looked for a difference in the evolution of these biomarkers within the two groups by running linear mixed models (one for each biomarker, package lme4) with a patient-level random effect. We considered that the evolutions were different between the two groups if the interaction term time * group was significant according to Wald test. All analyses were performed using SAS V.9.4 and R V.4.0.3 softwares.

## Ethical aspects

The study and its experimental protocol were approved by the AP-HP Scientific and Ethical Committee (IRB00011591 decision number CSE-210014). All data were fully anonymized before we accessed them, and the French law waived the requirement for informed consent for such data. Patients were informed that their EHR information could be reused after an anonymization process and those who objected to the reuse of their data were excluded. All methods were carried out in accordance with relevant guidelines (reference methodology MR-004 of the CNIL: Commission Nationale de l'Informatique et des Libertés). Authors had no access to any information that could identify individual participants during or after data collection.

## Results

### Population selection

A SLE diagnosis ICD-10 code was reported in the electronic health records of 4,533 patients admitted in one of the 39 university hospitals of the Greater Paris area, France between July 15, 2017, and February 9, 2022. Among them, 128 (2.8%) had an admission stay tagged with a COVID-19 diagnosis ICD-10 code between March 2020 and December 31, 2021. After individual review of the medical charts, 47 patients were excluded because 2019 EULAR/ACR classification criteria for SLE were not met (n = 22), SARS-CoV-2 infection occurred after December 2021 (n = 10) or was not proven (n = 7), data following COVID-19 infection were missing (n = 6), patients were younger than 16 (n = 1), or had a SLE diagnosed after COVID-19 (n = 1) (details provided in Fig 1).

Overall, 81 SLE patients (76 (96.2%) female, median [Q1-Q3] age 56.3 [40.6–68.3] years old) with COVID-19 fulfilling the selection criteria were included in the study. The characteristics of the COVID-19 in those patients are given Table 1. Around 30% (n = 25/81) of admissions for COVID-19 occurred before August 2020 S1 Fig. Eleven patients died during (n = 8) or shortly after (n = 3) discharge from the COVID-19 stay. Among COVID-19 survivors, the median [Q1-Q3] follow-up time after COVID-19 was of 289 [42–502] days.

### Matching procedure

We were able to match 79/81 (97.5%) COVID-19 patients to 79 unique unexposed SLE patients through our matching procedure. Despite matching, significant differences were still remained between groups regarding hydroxychloroquine (HCQ) treatment (51 (65.4%) SLE in the COVID group were receiving HCQ at index date as compared to 60 (75.9%) in the unexposed group; SMD = 0.251) and history of lupus nephritis (37 (48.1%) SLE patients in the COVID

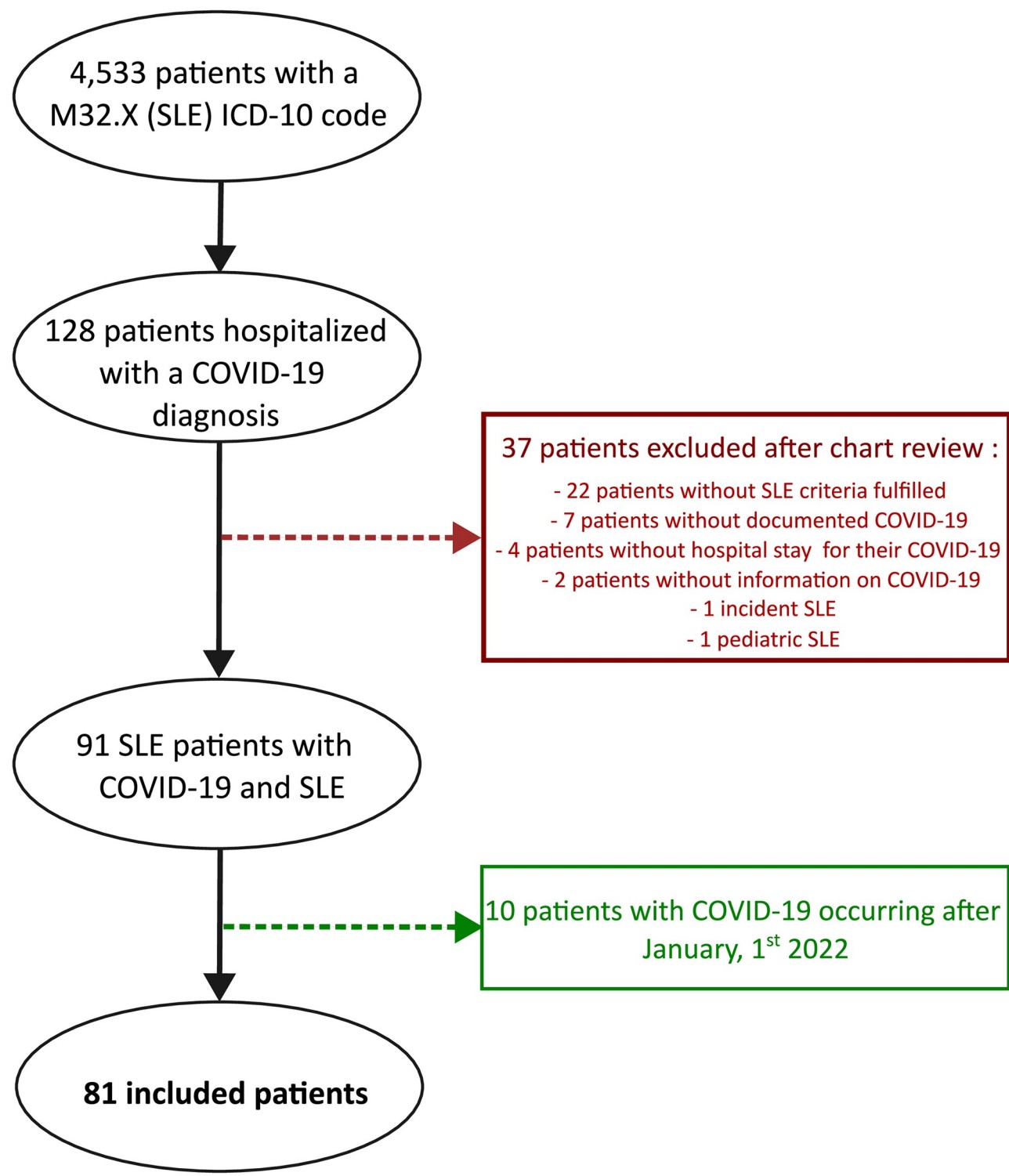

Fig 1. Selection of the exposed population.

**Table 1. Characteristics of the COVID-19 episode.**

| | SLE patients with COVID-19 n = 81 |
|---|---|
| Age, in years, median (Q1-Q3) | 56.4 [42.2–68.3] |
| Female gender, n (%) | 76 (93.8) |
| **COVID-19 episode** | |
| Prior anti-SARS-CoV2 vaccine, n (%) | 9 (11.1) |
| CT-severity score, (33 missing), n (%): | |
| *< 10%* | 8 (16.7) |
| *10–25%* | 17 (35.4) |
| *25–50%* | 12 (25.0) |
| *50–75%* | 9 (18.8) |
| *> 75%* | 2 (4.2) |
| **Treatment of COVID-19** | |
| Oxygen, n (%) | 54 (66.7) |
| More than > 6l.min$^{-1}$ of oxygen, n (%) | 31 (38.3) |
| Non-invasive mechanical ventilation*, n (%) | 18 (27.8) |
| Invasive mechanical ventilation, n (%) | 14 (17.5) |
| ECMO, n (%) | 1 (1.3) |
| Dexamethasone, n (%) | 23 (28.8) |
| Anti-cytokine mAb, n (%) | 5 (6.3) |
| Anti-SARS-CoV2 specific mAb, n (%) | 6 (7.6) |
| Anticoagulant treatment, n (%) | 27 (35.1) |
| Convalescent plasma therapy, n (%) | 2 (2.5) |
| Tapering of the IS treatment, n (%) | 18 (22.2) |
| **Outcome** | |
| Length of the hospital stay, in days (median [Q1-Q3] | 9.5 [6.0–20.0] |
| Thrombosis, n (%) | 6 (7.8) |
| Death during hospital stay, n (%) | 8 (9.8) |

Q1: first quartile; Q3: third quartile; ECMO: extracorporeal membrane oxygenation; mAb: monoclonal antiboy;

*non-invasive mechanical ventilation includes high flow oxygen therapy. IS: immunosuppressive

group had past history of lupus nephritis at index date as compared to 31 (39.7%) in the unexposed group; SMD = 0.153). The characteristics of the matched populations are given Table 2.

## Risk of SLE flare after COVID-19

During follow-up, 14 (17.7%) SLE patients from the COVID-19 group experienced a lupus flare as compared to 5 (6.3%) from the unexposed control group. Overall, lupus flare ups included lupus nephritis (n = 5, 4 in the COVID-19 group and 1 in the control group), severe immune cytopenia (n = 4, 3 in the COVID-19 group and 1 in the control group) and pleuritis (n = 3, 2 in the COVID-19 group and 1 in the control group). Flare were associated with a treatment modification in all cases: in 14 cases there was an increase in daily corticosteroid dosage, 2 patients had a reintroduction of hydroxychoroquine and 5 had an introduction of an immunosuppressive drug (mycophenolate, azathioprine or belimumab). Unadjusted HR showed that risk of lupus flare during a 6-month follow-up period were higher in SLE patients who experienced COVID-19 as compared to unexposed control (HR 3.62 [1.39–9.41]).

**Table 2. Characteristics of the matched populations.**

| | COVID-19 n = 79 | No COVID-19 n = 79 | Absolute SMD |
|---|---|---|---|
| Age, in years*, median (Q1-Q3) | 56.3 [40.6–68.3] | 55.9 [39.7–68.2] | 0.003 |
| Female gender*, n (%) | 76 (96.2) | 76 (96.2) | 0.000 |
| Afro-Caribbean ethnicity, n (%) | 30 (45.4) | 24 (36.4) | 0.185 |
| **Comorbidities** | | | |
| CKD*, n (%) | 22 (27.9) | 22 (27.9) | 0.000 |
| ESRD*, n (%) | 15 (18.9) | 15 (18.9) | 0.000 |
| **SLE disease** | | | |
| Years from SLE diagnosis, median [Q1-Q3] | 14.2 [5.6–22.3] | 11.8 [6.8–24.8] | 0.035 |
| History of lupus nephritis, n (%) | 37 (48.1) | 31 (39.7) | 0.153 |
| class III/IV nephritis, n (%) | 24 (31.2) | 22 (28.6) | 0.056 |
| Serological activity during the last 6 months*⁺, n (%) | | | 0.022 |
| Normal | 29 (36.7) | 30 (36.7) | |
| Abnormal | 35 (44.3) | 34 (43.0) | |
| Not performed | 15 (18.9) | 15 (18.9) | |
| **SLE flare in the last 6 months before index date** | 9 (11,4) | 15 (19,0) | -0.213 |
| SLE treatment at index date | | | |
| Hydroxychloroquine, n (%) | 51 (65.4) | 60 (75.9) | 0.251 |
| Steroids, n (%) | 53 (67.9) | 54 (68.3) | 0.027 |
| Prednisone equivalent daily dose, if any (mg/d) | 5 [5–9] | 7 [5–10] | 0.109 |
| IS drug, n (%) | 31 (39.2) | 32 (40.5) | 0.025 |
| Mycophenolate mofetil | 22 (28.2) | 20 (25.3) | |
| Azathioprine, | 5 (6.4) | 9 (11.4) | |
| Rituximab | 6(7.7) | 5(6.4) | |
| Recent modification of SLE treatment ‡, n (%) | 24 (30.4) | 23 (29.1) | 0.028 |
| **Follow-up** | | | |
| Death during follow-up, n (%) | 11 (14.0) | 3 (3.8) | |

Q1: first quartile; Q3: third quartile; CKD: Chronic kidney disease defined as an eGFR < 60mL/min and no end-stage renal disease; ESRD: End-stage renal disease defined as chronic dialysis or renal transplantation; IS: immunosuppressive; SMD: standardized mean differences.

Lupus nephritis classes refer to the ISN/RPSWG classification [17].

⁺C3 levels and anti-dsDNA IgG titers measured in the serum at the latest 6 months prior the index date were considered. When performed, serological activity was defined as either normal—when both C3 level and anti-dsDNA IgG titer were into the normal range–or abnormal–when C3 level was low and/or anti-dsDNA IgG titer was high.

* Matching variables.

‡ SLE treatment modification during the six months before the index date or during COVID episode.

After adjustment for HCQ use at index date and history of lupus nephritis, HR was 3.79 [1.49–9.65]. The median delay between the index date and lupus flare was shorter (47 [19–129] versus 77 [46–83] days) in the COVID-19 + group than in the COVID-19—group. The Kaplan-Meier curve of survival without lupus flare in the two matched groups is given Fig 2. The proportional hazard assumption was met for the whole period of analysis.

## Serological biomarkers for SLE activity after COVID19

During follow up, 85 SLE patients (46 COVID + and 39 COVID -) and 44 SLE patients positive for anti-dsDNA autoantibodies (17 COVID + and 27 COVID -) had at least one measurement of C3 levels and anti-dsDNA IgG titers, respectively. Using the mixed model approach to

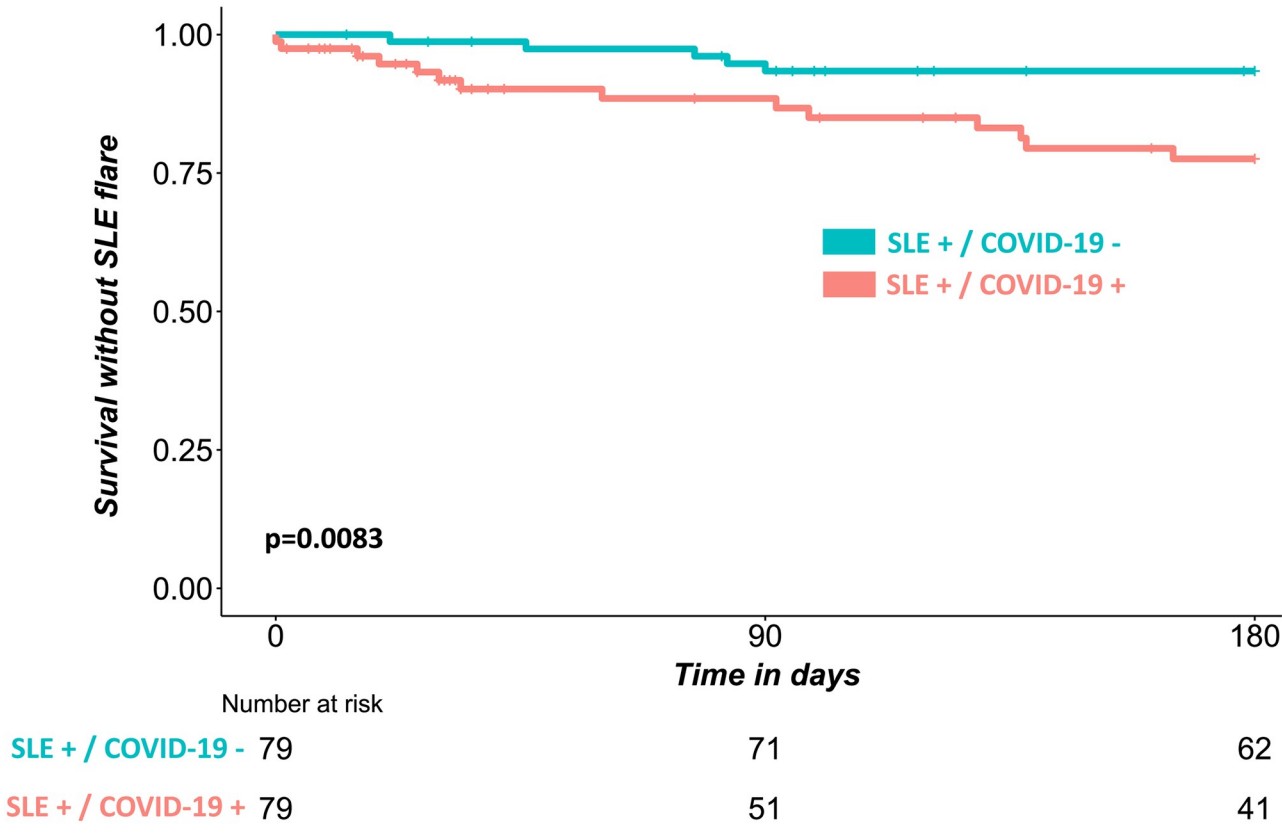

**Fig 2. Kaplan-Meier curves of the survival without flare among the matched populations.** P value was calculated with the log rank test.

analyze 184 measurements of C3 levels overtime, we observed that C3 levels decreased after the index date in COVID-19 patients and remained unchanged in the control groups (p = 0.085). Based on the analysis of 85 measurements, no difference was observed between groups regarding the evolution of anti-dsDNA IgG titers during follow-up. The fitted outcome of these immunological biomarkers in each group of interest is shown in Fig 3.

## Sensitivity analyses

We also observed unbalanced mortality between groups since 8 SLE patients died during the index stay in the COVID-19 group and none in the non-exposed group. To analyze the impact of competing risks, we performed a sensitivity analysis excluding the pairs (exposed and matched control) of exposed patients who died of COVID-19. The same proportion of events was observed in each group and the adjusted hazard ratio remained almost unchanged: 3.45 [1.35–8.81].

We also performed an analysis after extending the follow-up to 9 months after the index date. The number of events observed in both groups was the same as in the main analysis, and the adjusted HR was very similar: 3.79 [1.49–9.65].

## Risk factors for lupus flare after COVID-19

Factors associated with the occurrence of a lupus flare after COVID-19 were analyzed using competing risk models in the exposed populations. Death during follow-up was considered as

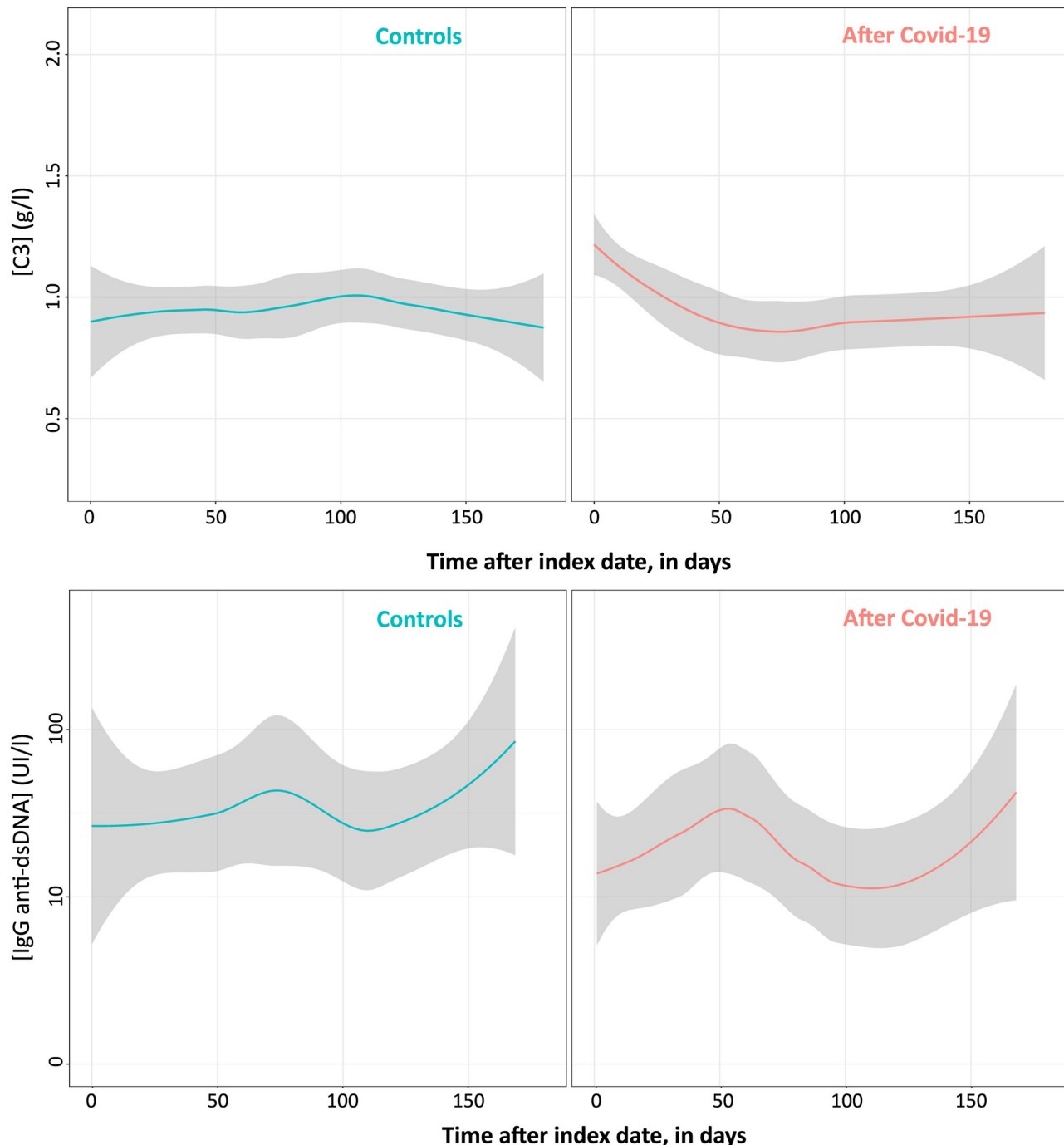

**Fig 3. C3 levels and anti-dsDNA IgG titers.** C3 levels after index date in the matched populations (upper panel). Anti-dsDNA IgG titers after index date among patients positive for anti-dsDNA autoantibodies (lower panel) in the matched populations. Lines represent the fitted LOESS model with its confidence interval. A log10 scale is used for the Y axis in the lower panel.

the competing event and factors associated with such risk were studied (Table 3). The only factor associated with the occurrence of a flare after COVID-19 was a prior anti-SARS-CoV2 vaccine with a HR of 4.00 [1.09–14.6]. Interestingly, tapering the SLE immunosuppressive drugs during COVID stay was not associated with an increased risk of flare (HR = 0.86 [0.23–3.18]). Results are given Table 3.

**Table 3. Factors associated with the occurrence of a flare or death during follow-up.**

|  | Risk of SLE flare | | Risk of death | |
|---|---|---|---|---|
|  | sdHR [95 CI] | p | sdHR [95 CI] | p |
| Age, in years | 0.98 [0.95–1.01] | 0.11 | 1.05 [1.02–1.09] | 0.003 |
| Male gender | 2.12 [0.22–20.3] | 0.51 | NA |  |
| **Comorbidities** |  |  |  |  |
| Chronic kidney disease | 0.66 [0.17–2.50] | 0.54 | 3.65 [1.06–12.6] | 0.041 |
| End-stage renal disease | 1.21 [0.32–4.64] | 0.78 | 0.98 [0.22–4.42] | 0.98 |
| **COVID-19** |  |  |  |  |
| > 6l.min-1 of oxygen during COVID | 0.24 [0.05–1.11] | 0.07 | 16.2 [2.05–129.1] | 0.008 |
| Prior anti-SARS-CoV2 vaccine | 4.00 [1.09–14.6] | 0.04 | NA |  |
| Dexamethasone treatment for COVID | 2.18 [0.76–6.29] | 0.15 | 1.85 [0.65–5.27] | 0.25 |
| **SLE disease** |  |  |  |  |
| Years from SLE diagnosis, | 0.97 [0.90–1.03] | 0.28 | 1.05 [1.01–1.09] | 0.019 |
| History of lupus nephritis | 0.60 [0.20–1.80] | 0.60 | 1.02 [0.30–3.45] | 0.98 |
| *class III/IV lupus nephritis* | 0.90 [0.27–3.02] | 0.86 | 1.45 [0.42–5.06] | 0.56 |
| Serological activity during the last 6 months[+] |  |  |  |  |
| *Normal* | ref | ref | ref | ref |
| *Abnormal* | 2.64 [0.73–9.54] | 0.14 | 0.53 [0.09–3.06] | 0.53 |
| *Not performed* | 0.59 [0.06–5.83] | 0.65 | 3.76 [0.95–14.9] | 0.06 |
| **SLE treatment** |  |  |  |  |
| Steroids at index date | 0.82 [0.28–2.41] | 0.73 | 1.92 [0.41–9.09] | 0.41 |
| IS drugs at index date | 0.38 [0.10–1.39] | 0.14 | 0.61 [0.16–2.36] | 0.47 |
| Tapering of IS drugs during COVID | 0.86 [0.23–3.18] | 0.86 | 0.75 [0.16–3.42] | 0.75 |

CKD: Chronic kidney disease defined as an eGFR < 60mL/min and no end-stage renal disease; ESRD: End-stage renal disease defined as chronic dialysis or renal transplantation; IS: immunosuppressive.

[+]C3 levels and anti-dsDNA IgG titers measured in the serum at the latest 6 months prior the index date were considered. When performed, serological activity was defined as either normal—when both C3 level and anti-dsDNA IgG titer were into the normal range–or abnormal–when C3 level was low and/or anti-dsDNA IgG titer was high.

sdHR = sub-distribution hazard ratio; 95CI: 95% confidence interval

## Discussion

Using the APHP Clinical Data Warehouse, we leveraged more than 4,000 electronic health records of SLE patients admitted in one of the 39 university hospitals of Paris area to investigate the link between COVID-19 and lupus flare. Our analysis demonstrated that SLE patients are at increased risk of lupus disease flare after COVID-19. Lupus flares were severe, including lupus nephritis in almost 30% of cases and occurreds shortly after COVID-19, suggesting a relationship between COVID-19 and flares. Our findings are consistent with the published evidence supporting a link between COVID-19 and lupus flares in case reports and short series [6, 9–11, 18, 19] and with a recent self-controlled case series study which observed an increased risk of flare after influenza infections [20]. Other viruses have been previously associated with the risk of SLE onset such as Epstein-Barr virus [21], parvovirus B19, human endogenous retroviruses or cytomegalovirus [22] but their role in the activity of the disease is less clear.

Type 1-interferon (IFN-1) play a key role in both SLE and COVID-19. IFN-1 is produced by plasmacytoid dendritic cells (pDCs) in response to SARS-CoV2 infection and contributes to the macrophage-induced cytokine storm (6) observed during COVID-19 episodes. IFN-1 and pDCs also play a central role in SLE, being involved in the pathogenesis and activity of the

disease (4). In such setting, the persistence of a low-grade inflammatory type 1-interferon activity in SLE patients who survive severe COVID-19 could play a role in subsequent lupus flares. The occurrence of various auto-antibodies following severe COVID-19 suggests a defect in tolerance mechanisms as a result of the rapid and exaggerated inflammatory responses to Sars-CoV-2 (14). Critically ill patients with severe infection caused by SARS-CoV-2 display intense extrafollicular B cell responses enriched in autoreactive potential similar to those previously reported in SLE [23, 24]. On one hand, uncensored extrafollicular expansion may be considered a dominant and adapted immune response that controls severe COVID-19 through acute inflammation; on the other hand, it may promote flare-ups in SLE patients.

Unexpectedly, the only significant risk factor for lupus flare following COVID-19 was a prior anti-SARS-CoV2 vaccine. This finding is clearly challenging–considering the demonstrated benefit of vaccination in SLE [25]. Severe SLE patients are usually treated with immunosuppressive drugs. Since anti-SarS-CoV-2 vaccination is highly recommended in patients treated with immunosuppressive (IS) drugs, vaccinated SLE patients may have a higher risk of relapse because they have more severe disease. Although such selection bias is plausible it should also be pointed out that 1) humoral response against SARS-CoV-2 following vaccination may be dramatically high in SLE patients with evidence of prior COVID-19 [26] and 2) SARS-CoV-2 vaccination was followed by increased *in vivo* production of IFNα by pDCs in SLE patients [27].

Our work has several limitations. First, although all SLE patients appeared to be followed on a regular basis regardless of COVID-19, our analysis, which is retrospective, might suffer from a surveillance bias where patients benefited from a reinforced follow-up after COVID-19. Additionally, some patients may have had symptoms related to a lupus flare but wrongly ascribed to incident COVID-19; such confusion appears very unlikely considering the long median delay that was observed between admission for COVID-19 and lupus flare of 47 [19–129] days. Third, the high mortality rate at the early phase of COVID-19 may have artificially inflated the hazard ratio due to informative censoring, despite our sensitivity analyses considering competing risk of death. Fourth, only SLE patients admitted for COVID-19 have been considered and our results might not be generalizable to mild or asymptomatic COVID-19 outpatients. Fifth, SLE being a rare disease and admission for COVID-19 a rather uncommon exposure, our study suffer from a relatively small sample size. The limited number of patients may have particularly affected the analysis of anti-dsDNA titers after the index date or the search for risk factors for flare that are both based on restricted subsets of patients that could be underpowered. Our study is however the largest on this matter and has sufficient power for significant results. Sixth, SLE activity acts probably as a confounding factor being associated with the risk of infection and the risk of flare and it may persist unmeasured residual differences in SLE activity between groups even if we used the serological activity as a matching variable. Because our study uses a "real-life" setting with data extracted from the electronic medical case records, we were not able to calculate traditional activity scores such as SLEDAI-2k because of missing variables, and we cannot be sure that the control group did not suffer from any mild covid infection not requiring hospitalization or outpatient visit. Last, we did not provide any mechanistic evidence of the link between COVID-19 and lupus flare.

Our study also has several strengths. We had access to a comprehensive clinical database that drew from electronic health records, detailed information on medical history, biology, treatments, procedures, and outcomes. Furthermore, we thoroughly examined the full text of medical reports that were generated by the medical staff on a daily basis as part of the standard of care. Additionally, our matching process, which took into account factors such as age, sex, lupus activity parameters (anti-dsDNA IgG titers and C3 levels), and organ damage (chronic kidney disease), helped mitigate the risk of selection bias. Lastly, we used a clinically relevant definition of flares based on the assessment of the patient by the treating physician.

In conclusion, SLE patients are at increased risk of lupus flare following symptomatic COVID-19. Short-term follow-up is warranted after hospital discharge.

## Supporting information

**S1 Fig. Temporal distribution of the COVID-19 episodes.**
(PDF)

**S2 Fig. Kaplan-Meier curve of the flare-free survival using 3 months and 12 months of follow-up.**
(PDF)

## Acknowledgments

The authors would like to thank the AP-HP data warehouse, which provided the data and the computing power to carry out this study under good conditions. We wish to thank all the medical colleges from the APHP departments of internal medicine, rheumatology, dermatology, and nephrology that gave their agreements for the use of the clinical data.

## Author Contributions

**Conceptualization:** Arthur Mageau, Christel Géradin, Kankoé Sallah, Thomas Papo, Karim Sacre, Jean-François Timsit.

**Data curation:** Arthur Mageau, Christel Géradin, Kankoé Sallah, Jean-François Timsit.

**Formal analysis:** Arthur Mageau, Jean-François Timsit.

**Funding acquisition:** Arthur Mageau, Karim Sacre.

**Investigation:** Arthur Mageau, Thomas Papo, Karim Sacre.

**Methodology:** Arthur Mageau, Kankoé Sallah, Thomas Papo, Karim Sacre, Jean-François Timsit.

**Project administration:** Arthur Mageau, Thomas Papo, Karim Sacre, Jean-François Timsit.

**Resources:** Christel Géradin, Thomas Papo, Jean-François Timsit.

**Software:** Arthur Mageau, Christel Géradin, Jean-François Timsit.

**Supervision:** Karim Sacre, Jean-François Timsit.

**Validation:** Arthur Mageau, Christel Géradin, Karim Sacre, Jean-François Timsit.

**Visualization:** Arthur Mageau.

**Writing – original draft:** Arthur Mageau, Christel Géradin, Karim Sacre.

**Writing – review & editing:** Arthur Mageau, Christel Géradin, Kankoé Sallah, Thomas Papo, Karim Sacre, Jean-François Timsit.

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
