## [Decision Letter · Decision Letter 0]

3 Nov 2023

PONE-D-23-32994Increased risk of systemic lupus erythematosus flare after admission for COVID-19: a multicenter matched cohort study coupling medico-administrative register and clinical electronic database.PLOS ONE

Dear Dr. Mageau,

Thank you for submitting your manuscript to PLOS ONE. After careful consideration, we feel that it has merit but does not fully meet PLOS ONE’s publication criteria as it currently stands. Therefore, we invite you to submit a revised version of the manuscript that addresses the points raised during the review process.

We look forward to receiving your revised manuscript.

Kind regards,

Rajendra Bhimma, PhD

Academic Editor

PLOS ONE

Journal Requirements:

1. ase ensure that your manuscript meets PLOS ONE's style requirements, including those for file naming. The PLOS ONE style templates can be found at 

https://journals.plos.org/plosone/s/file?id=ba62/PLOSOne_formatting_sample_title_authors_affiliations.pdf"

3. Thank you for stating the following financial disclosure: "Arthur Mageau was supported by a PhD fellowship provided by Fondation pour la Recherche Médicale (FDM202106013488). This work was supported by the Agence Nationale de la Recherche (n°ANR-21-COVR-0034 COVALUS), the University Paris Cité and the Assistance Publique Hôpitaux de Paris."

Please state what role the funders took in the study.  If the funders had no role, please state: The funders had no role in study design, data collection and analysis, decision to publish, or preparation of the manuscript."

5. Please amend either the title on the online submission form (via Edit Submission) or the title in the manuscript so that they are identical.

Reviewers' comments:

Reviewer's Responses to Questions

**Comments to the Author**

1. Is the manuscript technically sound, and do the data support the conclusions?

Reviewer #1: Partly

Reviewer #2: Partly

2. Has the statistical analysis been performed appropriately and rigorously? 

Reviewer #1: No

Reviewer #2: I Don't Know

3. Have the authors made all data underlying the findings in their manuscript fully available?

Reviewer #1: Yes

Reviewer #2: No

4. Is the manuscript presented in an intelligible fashion and written in standard English?

Reviewer #1: Yes

Reviewer #2: Yes

5. Review Comments to the Author

Reviewer #1: The authors tried to demonstrate the increased risk of systemic lupus erythematosus flare after admission for COVID-19 in a multicenter matched cohort study.

1. Design: the flare rate can be influenced by many factors, especially after an infection event. If the flare risk is increased after COVID-19 infection by a short period, such as 6 months in the manuscript, the rate would be back to baseline level thereafter. It is more convincing if the authors can present these data.

2. Besides from “tapering the IS”, changing prednisone could also be a common clinical practice after COVID-19 infection, how do you minimize the influence of this confounder ?

3. How do you choose matching factors? Such as chronic kidney disease, end-stage renal disease. Besides, CKD and ESRD were not independent factors.

4. When you define normal or abnormal “lupus biology”, which is not a common terminology, did you mean SACQ or SQCQ ?

5. Table 2: there are only 14 outcomes (flare events), however more than 10 factors were incorporated in the risk models. I am afraid that overfitting will occur.

Reviewer #2: This is a retrospective matched cohort study investigating the relationship between COVID-19 infection and subsequent SLE flare in patients with established SLE. The authors utilized a database with clinical and administrative data that included 11 million patients seen in 39 different hospitals in Paris, France. Patients with SLE and COVID-19 infection between March 2020 and December 2021 were identified by ICD10 code and verified by chart review. The outcome of interest was physician diagnosed SLE flare (with associated medication change) within the 6 months after COVID-19 infection. Exposed patients (i.e. those who had COVID-19 infection) were matched by age, gender, renal disease and lupus biology 1:1 with unexposed SLE patients and rates of SLE flare were compared. The authors found that patients who were hospitalized due to COVID-19 infection had a higher risk of SLE flare than matched unexposed patients (HR 3.79 after adjusting for differences in HCQ use and h/o lupus nephritis).

This is a well-designed matched cohort study, with the major limitation being surveillance bias in the exposed group. It is reasonable to assume that patients who were hospitalized would be more likely to connect with their providers and have lupus flares captured in the official record. This point is mentioned as point 4 in the limitations section of the manuscript but could be further emphasized. Other points as below.

- Why was six months chosen as the follow up time frame? Please provide a rationale. If the anti-interferon biology is the proposed mechanism for SLE flare, this seems like a long time for a flare to be potentially causally associated with the infection.

- Please include a sensitivity analysis with a 3 month follow up time frame for flares as this seems biologically more plausible

- Is there data available on what SLE medications hospitalized patients were discharged on? Or, data on how often immunosuppression was discontinued in these patients? This would be useful to know and could influence interpretation of the results (i.e. if patients who were hospitalized had their immunosuppressive medications discontinued it would not be surprising that they flared, and the flare would not be directly related to their COVID-19 infection).

- Details of COVID-19 infection in general are useful and would recommend including them in the main manuscript rather than supplemental materials; this will help readers understand the severity of the infection (i.e. 66% required oxygen, 25% had CT-severity scores of 25-50%).

- Why were two patients of the 81 unmatched? Please clarify in the manuscript.

- Why was C4 levels not included as a parameter of lupus biology?

- Please reword the conclusion in the abstract; as written it overstates causality, which cannot be demonstrated in this study design.

- On page 17, please remove line 357-358. It is anecdotal that a causal link is suggested by clinicians and a bit misleading.

- The discussion would benefit from further literature review and context, specifically the rates of SLE flares observed after viral infections that are not COVID19. There is only one reference on rates of flare after influenza infection. It is common wisdom that patients with SLE flare in the setting of viral infections (EBV, rhinovirus etc); how do rates of flare following COVID19 compare to rates with other viruses and why might they be different or the same?

6. PLOS authors have the option to publish the peer review history of their article (what does this mean?). If published, this will include your full peer review and any attached files.

Reviewer #1: No

Reviewer #2: No

---

## [Author Response · Author response to Decision Letter 0]

9 Dec 2023

Risk of systemic lupus erythematosus flare after COVID-19: a matched cohort study.

Revision #1

Point by point response to Reviewers’ comments.

Reviewer #1: The authors tried to demonstrate the increased risk of systemic lupus erythematosus flare after admission for COVID-19 in a multicentre matched cohort study.

1. Design: the flare rate can be influenced by many factors, especially after an infection event. If the flare risk is increased after COVID-19 infection by a short period, such as 6 months in the manuscript, the rate would be back to baseline level thereafter. It is more convincing if the authors can present these data.

Response: Thank you for this relevant point. We agree with Reviewer 1 that if our hypothesis is correct, the flare rate should return to baseline after a certain period. We chose 6 months of follow-up a priori because this would let us have enough follow-up for all the included patients and because patients with SLE have usually a visit with their physician at least once every 6 months. 

According to that comment, we looked at the risk of flare using 12 months of follow-up. You can see here the Kaplan-Meier curves of the flare-free survival. 

We observed that, according to our hypothesis, the flare rate of the COVID-19 negative patients goes back to baseline. We added this analysis in the electronic supplementary materials of the paper. 

2. Besides from “tapering the IS”, changing prednisone could also be a common clinical practice after COVID-19 infection, how do you minimize the influence of this confounder?

Response: As shown in the table S1 in the ESM, 23 (28.8%) of the COVID patients were treated with dexamethasone for their COVID episodes. Conversely, a decrease in the corticosteroid daily dose was considered as an “IS tapering”. As shown below, we did not find that a dexamethasone treatment or a tapering of the IS treatment was associated with the risk of flare using our competing risk analysis: 

 Risk of SLE flare

 sdHR [95 CI] p

Tapering of IS drugs during COVID 0.86 [0.23-3.18] 0.86

Dexamethasone treatment for COVID 2.18 [0.76-6.29] 0.15

This analysis has been added in table 2 in the manuscript.

3. How do you choose matching factors? Such as chronic kidney disease, end-stage renal disease. Besides, CKD and ESRD were not independent factors.

Response: This is of course a very important point. We wanted to use matching criteria related to the SLE “phenotype” and to the risk of flare. We chose age, sex and a marker of organ damage (CKD/ESRD) and a marker of SLE activity (dsDNA titer and C3 titer) because we believe that these variables match this definition and because they were available in our database. We agree that CKD and ESRD are not independent. We thought that renal transplantation and chronic haemodialysis really change the risk for a SLE patient to experience flare, we wanted to be sure that the number of patients with ESRD was similar within the two groups. Of note, patient with ESRD were not mixed with patients with CKD for the matching. We built a three-level variable: “No CKD/ESRD” / “CKD without ESRD” / “ESRD” and use this variable to perform the matching. 

4. When you define normal or abnormal “lupus biology”, which is not a common terminology, did you mean SACQ or SQCQ?

Response: Thank you for raising this point. In this variable, we looked only at the biological markers. Therefore, we think that using the SACQ/SQCQ terminology would not be very relevant. However, and according to your comment, we modified the term “lupus biology” to the one of “serological activity” throughout the manuscript. 

5. Table 2: there are only 14 outcomes (flare events), however more than 10 factors were incorporated in the risk models. I am afraid that overfitting will occur.

Response: Indeed, we observed only 14 flare events during follow-up. However, we did not include 10 factors in the risk models. In the main model (i.e. the marginal Cox proportional hazard model), we have included only three variables: the exposure group (“COVID”/”no COVID”, the use of hydroxychloroquine at baseline and a history of lupus nephritis. And, for the competing risk models, we ran only univariate analyses. 

Reviewer #2: This is a retrospective matched cohort study investigating the relationship between COVID-19 infection and subsequent SLE flare in patients with established SLE. The authors utilized a database with clinical and administrative data that included 11 million patients seen in 39 different hospitals in Paris, France. Patients with SLE and COVID-19 infection between March 2020 and December 2021 were identified by ICD10 code and verified by chart review. The outcome of interest was physician diagnosed SLE flare (with associated medication change) within the 6 months after COVID-19 infection. Exposed patients (i.e. those who had COVID-19 infection) were matched by age, gender, renal disease and lupus biology 1:1 with unexposed SLE patients and rates of SLE flare were compared. The authors found that patients who were hospitalized due to COVID-19 infection had a higher risk of SLE flare than matched unexposed patients (HR 3.79 after adjusting for differences in HCQ use and h/o lupus nephritis).

This is a well-designed matched cohort study, with the major limitation being surveillance bias in the exposed group. It is reasonable to assume that patients who were hospitalized would be more likely to connect with their providers and have lupus flares captured in the official record. This point is mentioned as point 4 in the limitations section of the manuscript but could be further emphasized. Other points as below.

Response: Thank you for your positive comment. We agree, as we already acknowledged that the risk of a surveillance bias is one of the main limitations of the study. According to this query, we modified the discussion section to put this limitation as the first one. However, we would like to point out that, we have all the tertiary, academic medical centres of the Paris area in our database, and that we can reasonably assume that almost all patients with SLE are followed in these centres. Then, it seems quite unlikely that we could have “missed” a flare. 

- Why was six months chosen as the follow up time frame? Please provide a rationale. If the anti-interferon biology is the proposed mechanism for SLE flare, this seems like a long time for a flare to be potentially causally associated with the infection.

- Please include a sensitivity analysis with a 3 month follow up time frame for flares as this seems biologically more plausible

Response: Thank you for this interesting point. Reviewer #1 also raised commented on the length of the follow-up. We chose 6 months of follow-up a priori because this would let us have enough follow-up for all the included patients and because patients with SLE have usually a visit with their physician at least once every 6 months. However, we don’t know through which mechanisms a viral infection could trigger a flare and therefore after which delay the maximum risk of flare after COVID should be expected. According to that comment, we ran the analysis with a global censoring after 3 months. Here is the Kpaln-Meier curve: 

Using this length of follow-up, we have calculated a hazard ratio of 2.45 [0.73-8.23] that failed to reach significance (p=0.15) using the same model that the one of the main analysis. We have included this analysis in the electronic supplementary materials of the manuscript. 

- Is there data available on what SLE medications hospitalized patients were discharged on? Or, data on how often immunosuppression was discontinued in these patients? This would be useful to know and could influence interpretation of the results (i.e. if patients who were hospitalized had their immunosuppressive medications discontinued it would not be surprising that they flared, and the flare would not be directly related to their COVID-19 infection).

Response: Of course, we agree that the modification of the immunosuppressive treatment during the COVID-19 episode could be a factor explaining at least a part of the risk for a post-COVID flare. We added the number of patients with a tapering of their immunosuppressive treatment during the viral infection in the table summarizing the characteristics of the COVID episode. We have already studied this parameter in our analysis of the factor influencing the risk of flare after a COVID-19 episode in SLE patients (Table 3 of the main manuscript). The sdHR for the risk of lare associated with this variable was 0.86 [0.23-3.18] (p=0.86). Therefore, we did not observe any significant effect of a decrease in the immunosuppressive treatment during the COVID episode. However, we had only 14 flare events so it is likely that we could be underpowered in this analysis. We added a sentence in the discussion section to better underline this :” The limited number of patients may have particularly affected the analysis of anti-dsDNA titers after the index date or the search for risk factors for flare that are both based on restricted subsets of patients that could be underpowered.”. 

- Details of COVID-19 infection in general are useful and would recommend including them in the main manuscript rather than supplemental materials; this will help readers understand the severity of the infection (i.e. 66% required oxygen, 25% had CT-severity scores of 25-50%).

Response: According to that comment, we added the table summarizing the characteristics of the COVID-19 infection in the main manuscript. 

- Why were two patients of the 81 unmatched? Please clarify in the manuscript.

Response: Two COVID patients remained unmatched because they were very particular (young male with ESRD), so they did not have any similar counterpart in the non-exposed pool. We clarified that in the manuscript. 

- Why was C4 levels not included as a parameter of lupus biology?

Response: Thank you for raising this point. We included only C3 levels and not C4 because C3 is usually considered as a more informative marker of the lupus serological activity (Semin Arthritis Rheum 2020 Oct;50(5):1081-1086.doi: 10.1016/j.semarthrit.2020.06.009.). Since complement level was included in the matching procedure, we wanted to use only one of its components to decrease the risk of missing or contradictory values. 

- Please reword the conclusion in the abstract; as written it overstates causality, which cannot be demonstrated in this study design.

Response: According to this comment, we have rephrased the sentence in the conclusion of the abstract to: “COVID-19 is associated with an increased risk of flare in SLE.” 

- On page 17, please remove line 357-358. It is anecdotal that a causal link is suggested by clinicians and a bit misleading.

Response: According to this comment, we removed the problematic sentence. 

- The discussion would benefit from further literature review and context, specifically the rates of SLE flares observed after viral infections that are not COVID19. There is only one reference on rates of flare after influenza infection. It is common wisdom that patients with SLE flare in the setting of viral infections (EBV, rhinovirus etc); how do rates of flare following COVID19 compare to rates with other viruses and why might they be different or the same?

Response: We agree on the fact that the discussion would benefit from further literature review. We have added several new references in the discussion section, and especially studies relative to the risk of post viral flare.

---

## [Decision Letter · Decision Letter 1]

14 May 2024

PONE-D-23-32994R1Risk of systemic lupus erythematosus flare after COVID-19: a matched cohort study.PLOS ONE

Dear Dr. Mageau,

Thank you for submitting your manuscript to PLOS ONE. After careful consideration, we feel that it has merit but does not fully meet PLOS ONE’s publication criteria as it currently stands. Therefore, we invite you to submit a revised version of the manuscript that addresses the points raised during the review process.

We look forward to receiving your revised manuscript.

Kind regards,

Rajendra Bhimma, PhD

Academic Editor

PLOS ONE

Additional Editor Comments:

See comments from reviewers

Reviewers' comments:

Reviewer's Responses to Questions

**Comments to the Author**

1. If the authors have adequately addressed your comments raised in a previous round of review and you feel that this manuscript is now acceptable for publication, you may indicate that here to bypass the “Comments to the Author” section, enter your conflict of interest statement in the “Confidential to Editor” section, and submit your "Accept" recommendation.

Reviewer #3: (No Response)

Reviewer #4: All comments have been addressed

Reviewer #5: All comments have been addressed

2. Is the manuscript technically sound, and do the data support the conclusions?

Reviewer #3: Partly

Reviewer #4: Yes

Reviewer #5: Yes

3. Has the statistical analysis been performed appropriately and rigorously? 

Reviewer #3: N/A

Reviewer #4: Yes

Reviewer #5: Yes

4. Have the authors made all data underlying the findings in their manuscript fully available?

Reviewer #3: Yes

Reviewer #4: Yes

Reviewer #5: No

5. Is the manuscript presented in an intelligible fashion and written in standard English?

Reviewer #3: Yes

Reviewer #4: Yes

Reviewer #5: Yes

6. Review Comments to the Author

Reviewer #3: The authors submitted a matched cohort study to assess the risk for flare in SLE patients during 6 months after COVID-19 infection by using the Assistance Publique - Hôpitaux de Paris Clinical Data Warehouse. This topic is very interesting since only case reports or case series have been published so far. Conversely, literature is lacking on large studies specifically designed to assess the risk of flare in SLE patients. The study design included the enrollment of SLE patients hospitalized with a COVID-19 diagnosis code (March 2020 - December 2021) matched to one SLE control patient with an exact matching procedure using age ±3 years, gender, chronic kidney disease, end-stage renal disease, and serological activity. Indeed, starting from 4,533 SLE patients retrieved from the database, a small number of patients (N=81) have been admitted for COVID-19 and then compared to 79 matched unexposed subjects. A flare occurred in 17.7% of evaluated patients, with a risk of flare higher in exposed SLE patients (HR 3.79).

Despite the interest aroused by the study, I have some comments for the authors.

1) The retrospective design and the small sample size certainly represent a limitation that should be underlined.

2) One important limit that should be discussed is certainly the definition of flare. Indeed, several definition has been proposed for flare, generally according to a specific disease activity index, i.e SLEDAI-2k. A punctual definition can ensure homogeneity of the data, particularly when multicenter studies are involved. In the present study the occurrence of flare after COVID-19 infection is the main outcome; however, the definition of flare is not very accurate, since was based on physician judgement or on change in SLE treatment. Indeed, this aspect represents an important limitation for the study, and it should be discussed. Furthermore, the change in SLE treatment that led to the definition of flare should be accurately described.

3) What was the disease activity of the patients at the time of COVID19 infection? This aspect could influence the risk for infection and deserves to be further explored or discussed.

4) It is plausible that the change in SLE treatment after COVID infection could facilitate a disease flare. This aspect has been added in the second version of the manuscript, but it should be interesting to provide information about the glucocorticoids dosage and the specific immunosuppressant taken at the time of infections.

5) The authors described a higher prevalence of history of nephritis in exposed than unexposed patients; however, no differences were found among the two groups in terms of treatment. So, we assume that the nephritis was not active at the time of infection. Is it correct?

6) Lupus nephritis accounted for the flare in 30% of cases. Were these flare-ups of patients who already had renal involvement or new onset? Please clarify this aspect.

Reviewer #4: This is a matched cohort study (retrospective) assessing the risk of systemic lupus erythematosus flare after COVID-19. The authors have answered the comments and revised their limitations

Reviewer #5: I congratulate the authors for addressing the majority of the comments from the previous review. I would like to declare that I did not participate in the previous round of review.

I only would like to echo a couple of comments from the previous round with some suggestions to consider:

First, I concur with comment #2 of reviewer #2. I wonder the reason for adjusting the Cox models for HCQ use "at index." Considering that the authors declared having access to the full electronic medical record, I would suggest adjusting for HCQ and IS use "at discharge." I consider this approach will be more robust than the exposure that could have changed by the time of the flare occurrence.

Second, Surveillance bias (already acknowledged) might be tackled by comparing the number of visits/encounters between the groups.

Minor comments:

- I recommend avoiding language of 'causal relationship' used in the discussion. The authors did recognize several limitations that indeed limit this conclusion.

- Consider using 'severe COVID-19 infection' throughout the manuscript to clarify that the study population were patients 'hospitalized.'

For the 'flare' definition, I would guess that the change in treatment refers to the increase in dose or number of medications (and not the opposite). A brief clarification would be helpful.

7. PLOS authors have the option to publish the peer review history of their article (what does this mean?). If published, this will include your full peer review and any attached files.

Reviewer #3: No

Reviewer #4: No

Reviewer #5: No

---

## [Author Response · Author response to Decision Letter 1]

30 May 2024

Point by point response to Reviewers’ comments.

Reviewer #3: 

The authors submitted a matched cohort study to assess the risk for flare in SLE patients during 6 months after COVID-19 infection by using the Assistance Publique - Hôpitaux de Paris Clinical Data Warehouse. This topic is very interesting since only case reports or case series have been published so far. Conversely, literature is lacking on large studies specifically designed to assess the risk of flare in SLE patients. The study design included the enrolment of SLE patients hospitalized with a COVID-19 diagnosis code (March 2020 - December 2021) matched to one SLE control patient with an exact matching procedure using age ±3 years, gender, chronic kidney disease, end-stage renal disease, and serological activity. Indeed, starting from 4,533 SLE patients retrieved from the database, a small number of patients (N=81) have been admitted for COVID-19 and then compared to 79 matched unexposed subjects. A flare occurred in 17.7% of evaluated patients, with a risk of flare higher in exposed SLE patients (HR 3.79).

Despite the interest aroused by the study, I have some comments for the authors.

The retrospective design and the small sample size certainly represent a limitation that should be underlined.

Response: We totally agree with Reviewer #3 on the fact that the small sample size and the retrospective design represent a limitation. We modified the discussion section in the manuscript to better underline that. However, we would like to point out that although the study uses a retrospective design, the data are collected prospectively in the APHP data warehouse. Besides, we believe that we had enough power to conclude on a difference between exposed and non-exposed patients since we observed a significant difference in the main analysis. 

2) One important limit that should be discussed is certainly the definition of flare. Indeed, several definitions has been proposed for flare, generally according to a specific disease activity index, i.e SLEDAI-2k. A punctual definition can ensure homogeneity of the data, particularly when multicenter studies are involved. In the present study the occurrence of flare after COVID-19 infection is the main outcome; however, the definition of flare is not very accurate, since was based on physician judgement or on change in SLE treatment. Indeed, this aspect represents an important limitation for the study, and it should be discussed. Furthermore, the change in SLE treatment that led to the definition of flare should be accurately described.

Response: This is indeed a very important point. SLE is a very heterogeneous disease and we agree that several definitions of flare have been described in the literature. However, we rather think that our flare definition is one of the strengths of our work. Our definition is clinically relevant because we used physician's assessment or an increase in the immunosuppressive treatment as a proxy for flare. Therefore, we believe that we get the meaningful flares, even if we might have missed mild or asymptomatic ones. Our study is in “real-life” setting with data extracted from the medical case records. Therefore, we would have a lot of missing data regarding the SLEDAI-2k variables that may not have been described by the physician in his report. We added a sentence in the discussion to acknowledge that. 

Regarding the change in SLE treatment, we added a sentence in the results section to describe the change in SLE treatment in case of a flare. 

3) What was the disease activity of the patients at the time of COVID19 infection? This aspect could influence the risk for infection and deserves to be further explored or discussed.

Response: Unfortunately, as previously stated, we were not able to calculate activity scores such as SLEDAI-2k at index date for all patients. We provided in Table 2 several variables associated with SLE activity at index time: the serological activity, the amount of daily steroid, and a recent modification in SLE treatment. To address reviewer’s query, we added one new variable associated with SLE activity in this table: the number of patients with a flare within the last 6 months before index date. There were 9 patients with a flare in the last 6 months in the COVID-19 group versus 15 in the non-exposed group. We used serological activity as a matching variable in order to make sure that disease activity is well balanced between the groups. However, it is true that unmeasured residual differences in SLE activity between groups may persist. According to reviewer’s comment, we added a sentence in the discussion to acknowledge that SLE activity could influence the risk of infection and the risk of flare, acting then as a confounding factor. 

4) It is plausible that the change in SLE treatment after COVID infection could facilitate a disease flare. This aspect has been added in the second version of the manuscript, but it should be interesting to provide information about the glucocorticoids dosage and the specific immunosuppressant taken at the time of infections.

Response: We agree that this is a very important information. Details on the dosage of glucocorticoids at the time of infections are given in Table 2. 

5) The authors described a higher prevalence of history of nephritis in exposed than unexposed patients; however, no differences were found among the two groups in terms of treatment. So, we assume that the nephritis was not active at the time of infection. Is it correct?

Response: Yes, it is correct. Nephritis was not active at the time of infection for all patients. 

6) Lupus nephritis accounted for the flare in 30% of cases. Were these flare-ups of patients who already had renal involvement or new onset? Please clarify this aspect.

Response: We have represented below the repartition of post-index renal flares according to the history lupus nephritis: 

 No history of renal flare n=90 History of renal flare n=68

Lupus nephritis (all classes) 1 (1.1%) 4 (5.9%)

Class III or IV LN 0 2 (2.9%)

Indeed, it is true that most of the patients with post-index lupus nephritis already had renal involvement. 

Because of these numbers are really small, we would prefer not to give these details in the manuscript. 

Reviewer #4: 

This is a matched cohort study (retrospective) assessing the risk of systemic lupus erythematosus flare after COVID-19. The authors have answered the comments and revised their limitations.

Response: Thank you for your comments that helped us to improve our manuscript.

Reviewer #5: 

I congratulate the authors for addressing the majority of the comments from the previous review. I would like to declare that I did not participate in the previous round of review.

Response: Thank you for your positive comment. 

I only would like to echo a couple of comments from the previous round with some suggestions to consider:

First, I concur with comment #2 of reviewer #2. I wonder the reason for adjusting the Cox models for HCQ use "at index." Considering that the authors declared having access to the full electronic medical record, I would suggest adjusting for HCQ and IS use "at discharge." I consider this approach will be more robust than the exposure that could have changed by the time of the flare occurrence.

Response: Thank you for raising this point. We adjusted for HCQ or IS use as baseline variables because index time is the beginning of the follow-up. At first, we thought about using time of discharge as time 0 but we realized that it would create an immortal time bias during the time between index and discharge. It would have been quite problematic knowing that some SLE flares occurred before discharge. Nevertheless, during the sensitivy analysis, we changed our time 0, using discharge at time 0 and excluding patients dying during the COVID-19 hospitalization. The results of this sensitivity analysis are already in the manuscript and are similar to the main one. Unfortunately, our data are not granular enough to let us use treatment exposure as a time-varying variable in the Cox model. 

In addition, most of the flares that we observed happened in a short time after COVID-19 episode (median [IQR] of 47 [19-129] days between COVID and flare. It is therefore likely that most patients' treatment was stable during this time. 

Second, Surveillance bias (already acknowledged) might be tackled by comparing the number of visits/encounters between the groups.

Response: Thank you for raising this point. We had the same idea during the study process. However, most of our study took place during the pandemic. During that period, patient’s follow-ups were quite disorganized and we were afraid that the number of visits/encounters would not reflect accurately the actual number of contacts between patients and their physician during that time. 

Minor comments:

- I recommend avoiding language of 'causal relationship' used in the discussion. The authors did recognize several limitations that indeed limit this conclusion.

Response: According to reviewer’s comment, we removed the term “causal relationship” from the manuscript. 

- Consider using 'severe COVID-19 infection' throughout the manuscript to clarify that the study population were patients 'hospitalized.'

Response: We would prefer not to use severe COVID-19 infection throughout the manuscript because, although most of the patients had severe COVID-19 with a high mortality rate, our inclusion criterion was “admitted in hospital with COVID-19 as the main reason for hospitalisation” and not “severe COVID-19”. Therefore, some patients in the COVID-19 group are heavily immunocompromised patients who have benne hospitalized to receive an anti-SARS-CoV-2 treatment during a symptomatic but not severe COVID-19 episode. 

For the 'flare' definition, I would guess that the change in treatment refers to the increase in dose or number of medications (and not the opposite). A brief clarification would be helpful.

Response: Thank you for this relevant point. We have modified the manuscript to clarify this point: “A SLE flare was defined if was 1) considered by the physician in the medical record and ii) followed by an increase in dose or number of medications given for SLE in the setting of care.” (Methods section)

---

## [Decision Letter · Decision Letter 2]

3 Jul 2024

PONE-D-23-32994R2Risk of systemic lupus erythematosus flare after COVID-19: a matched cohort study.PLOS ONE

Dear Dr. Mageau,

Thank you for submitting your manuscript to PLOS ONE. After careful consideration, we feel that it has merit but does not fully meet PLOS ONE’s publication criteria as it currently stands. Therefore, we invite you to submit a revised version of the manuscript that addresses the points raised during the review process.

We look forward to receiving your revised manuscript.

Kind regards,

Sham Santhanam

Academic Editor

PLOS ONE

Journal Requirements:

**Additional Editor Comments:**

Please revise the manuscript in accordance with the reviewers comments.

Reviewers' comments:

Reviewer's Responses to Questions

**Comments to the Author**

1. If the authors have adequately addressed your comments raised in a previous round of review and you feel that this manuscript is now acceptable for publication, you may indicate that here to bypass the “Comments to the Author” section, enter your conflict of interest statement in the “Confidential to Editor” section, and submit your "Accept" recommendation.

Reviewer #4: All comments have been addressed

Reviewer #6: (No Response)

2. Is the manuscript technically sound, and do the data support the conclusions?

Reviewer #4: Yes

Reviewer #6: Yes

3. Has the statistical analysis been performed appropriately and rigorously? 

Reviewer #4: Yes

Reviewer #6: Yes

4. Have the authors made all data underlying the findings in their manuscript fully available?

Reviewer #4: Yes

Reviewer #6: Yes

5. Is the manuscript presented in an intelligible fashion and written in standard English?

Reviewer #4: Yes

Reviewer #6: Yes

6. Review Comments to the Author

Reviewer #4: The authors have addressed the comments in the revised version, in this matched cohort study assessing risk of SLE flare after COVID.

Reviewer #6: The premise of study is intriguing. Most of the necessary corrections have already been made by previous reviewers. Although, I was not part of previous reviews, I have just one major comment to make. Since there was no patient communication involved in the study and everything was obtained from health records, how could the investigators ensure that the control group did not suffer from any mild covid infection not requiring hospitalization or OPD consult? Majority of COVID-19 patients improved with OTC treatment. This seems like a very significant flaw in the methodology that makes the results of the study unreliable. This should be included in the discussion.

Also, the study only involves participants who were hospitalized. The results cannot be generalized for all covid exposed SLE patients. The title of the study should be changed to ‘Risk of SLE flare after Covid-19 hospitalization’ and same correction should also be made in conclusion in abstract.

7. PLOS authors have the option to publish the peer review history of their article (what does this mean?). If published, this will include your full peer review and any attached files.

Reviewer #4: No

Reviewer #6: **Yes: **Kushagra Gupta, MD

---

## [Author Response · Author response to Decision Letter 2]

5 Jul 2024

Risk of systemic lupus erythematosus flare after COVID-19 hospitalization: a matched cohort study.

Revision #3

Point by point response to Reviewers’ comments.

Reviewer #4: 

The authors have addressed the comments in the revised version, in this matched cohort study assessing risk of SLE flare after COVID.

Response: Thank you for your comments that helped us to improve our manuscript.

Reviewer #6: 

The premise of study is intriguing. Most of the necessary corrections have already been made by previous reviewers. 

Response: Thank you for your positive comment.

I have just one major comment to make. Since there was no patient communication involved in the study and everything was obtained from health records, how could the investigators ensure that the control group did not suffer from any mild covid infection not requiring hospitalization or OPD consult? Majority of COVID-19 patients improved with OTC treatment. This seems like a very significant flaw in the methodology that makes the results of the study unreliable. This should be included in the discussion.

Response: This is indeed a very relevant point. We agree that even if we are sure to have all the COVID-19 hospitalizations, we cannot be sure that the control group did not suffer from any mild COVID-19. However, we disagree on the fact that this is a very significant flaw since we clearly defined the exposure. We considered symptomatic, hospitalized form of COVID-19 as out exposure of interest and not any COVID-19. Therefore, our conclusion is that a short-term follow-up is warranted after hospital discharge for SLE patients hospitalized for COVID-19. Our hypothesis is that it is the exaggerated immune response seen in severe COVID-19 that may be responsible for the risk of SLE flares.

According to Reviewer’s comment, we have added a sentence in the discussion section to better underline that: “we cannot be sure that the control group did not suffer from any mild covid infection not requiring hospitalization or outpatient visit.” 

Also, the study only involves participants who were hospitalized. The results cannot be generalized for all covid exposed SLE patients. The title of the study should be changed to ‘Risk of SLE flare after Covid-19 hospitalization’ and same correction should also be made in conclusion in abstract.

Response: We agree with Reviewer#6 on that point. We have modified the title and the abstract of the paper consequently.

---

## [Decision Letter · Decision Letter 3]

9 Aug 2024

Risk of systemic lupus erythematosus flare after COVID-19 hospitalization: a matched cohort study.

PONE-D-23-32994R3

Dear Dr. ARTHUR MAGEAU,

We’re pleased to inform you that your manuscript has been judged scientifically suitable for publication and will be formally accepted for publication once it meets all outstanding technical requirements.

Kind regards,

Sham Santhanam

Academic Editor

PLOS ONE

Additional Editor Comments (optional):

Reviewers' comments:

Reviewer's Responses to Questions

**Comments to the Author**

1. If the authors have adequately addressed your comments raised in a previous round of review and you feel that this manuscript is now acceptable for publication, you may indicate that here to bypass the “Comments to the Author” section, enter your conflict of interest statement in the “Confidential to Editor” section, and submit your "Accept" recommendation.

Reviewer #6: All comments have been addressed

Reviewer #7: All comments have been addressed

2. Is the manuscript technically sound, and do the data support the conclusions?

Reviewer #6: Yes

Reviewer #7: Yes

3. Has the statistical analysis been performed appropriately and rigorously? 

Reviewer #6: Yes

Reviewer #7: I Don't Know

4. Have the authors made all data underlying the findings in their manuscript fully available?

Reviewer #6: Yes

Reviewer #7: No

5. Is the manuscript presented in an intelligible fashion and written in standard English?

Reviewer #6: Yes

Reviewer #7: Yes

6. Review Comments to the Author

Reviewer #6: Thank you for addressing all queries. I have no more queries to raise.

All the best for your study!

Reviewer #7: The authors have addressed all the reviewer comments and the corrections made are satisfactory.

Best wishes to the authors for their work.

7. PLOS authors have the option to publish the peer review history of their article (what does this mean?). If published, this will include your full peer review and any attached files.

Reviewer #6: No

Reviewer #7: No

---

## [Editor Report · Acceptance letter]

14 Aug 2024

PONE-D-23-32994R3 

PLOS ONE

Dear Dr. Mageau, 

I'm pleased to inform you that your manuscript has been deemed suitable for publication in PLOS ONE. Congratulations! Your manuscript is now being handed over to our production team.

Kind regards, 

on behalf of

Dr. Sham Santhanam 

Academic Editor

PLOS ONE